# Putting the Gaming Experience at the Center of the Therapy—The Video Game Therapy^®^ Approach

**DOI:** 10.3390/healthcare11121767

**Published:** 2023-06-15

**Authors:** Francesco Bocci, Ambra Ferrari, Marcello Sarini

**Affiliations:** 1Playability Association, Via Sersane, 56, 25050 Ome, Italy; fbocci80@gmail.com (F.B.); ambraferrari@gmail.com (A.F.); 2Game Science Research Center, Via San Ponziano, 6, 55100 Lucca, Italy; 3Department of Psychology, University of Milano, Bicocca Piazza Ateneo Nuovo 1, 20126 Milano, Italy

**Keywords:** videogame, therapy, psychology, flow, individual psychoanalytic therapy, creative couple

## Abstract

**Simple Summary:**

Genesis and definition of the VGT^®^ approach.

**Abstract:**

Video games have been increasingly used as a form of therapy for various mental health conditions. Research has shown that video games can be used to treat conditions such as depression, anxiety, PTSD, and addiction. One of the main benefits of video games in therapy is that they can provide a sense of engagement and immersion that traditional therapy methods may lack. Additionally, video games can teach valuable skills such as problem solving, decision making, and coping strategies. Video games can also simulate real-life scenarios, allowing individuals to practice and improve social skills in a safe and controlled environment. Furthermore, video games can provide feedback and track progress objectively and quantifiably. This paper proposes an approach, the Video Game Therapy^®^ (VGT^®^) approach, where game experience is put at the center of the therapy in a tailored way, connecting the individual patient’s personality, the therapy’s goals, and the suggested type of video game through the Myers Briggs Type Indicator (MBTI).VGT^®^’s core assumption is that playing video games could facilitate patients in reaching conditions where traditional methodologies and therapeutic approaches could work best. VGT^®^ was elaborated according to the Adlerian therapy vision and, consequently, the different phases of Adlerian therapy and VGT^®^ match. Despite the use of video games in psychotherapy might have some adverse effects in specific cases, VGT^®^ is currently used in three associations with positive results in promoting emotional experimentation and literacy, social feeling, sense of identity, and activating cognitive processes. Future developments include expanding the use of VGT^®^ further to validate such results from a statistical point of view.

## 1. Introduction

A video game is an electronic game that can be played on a computer or a gaming console. It typically involves a user interacting with a visual display, such as a television or computer screen, to control characters or objects within a virtual environment.

There is no consensus in the literature about the definition of a game [1]. Philosophers and game designers have provided various definitions of what constitutes a “game”. Dutch historian Johan Huizinga defined a game as “a free activity standing quite consciously outside ‘ordinary’ life as being ‘not serious’, but at the same time absorbing the player intensely and utterly” [2]. Game designer Bernard Suits described a game as “the voluntary attempt to overcome unnecessary obstacles” [3]. Game theorists Katie Salen and Eric Zimmerman defined a game as “a system in which players engage in an artificial conflict, defined by rules, that results in a quantifiable outcome” [4]. Game designer Chris Crawford defined a game as “a problem-solving activity, approached with a playful attitude” [5]. All those different definitions emphasize some peculiar characteristics of (video)games that could be useful when considering a gaming experience as a therapeutic experience. A non-exhaustive list would include:Engagement: Video games are designed to be engaging and immersive, which can make them effective at keeping individuals focused and motivated during therapy sessions.Interactivity: Video games are interactive, allowing individuals to actively participate in therapy. This can be especially beneficial for individuals with difficulty in verbalizing their thoughts and feelings.Visual and auditory feedback: Video games can provide visual and auditory feedback, which can be used to reinforce positive behaviors and provide instant feedback on progress.Customization: Video games can be customized to suit the specific needs and goals of the individual. This can include tailoring the difficulty level, the type of game, and the session length.Variety: Video games offer a wide range of options, from action games to puzzles and simulations, which can help individuals find a type of game that they enjoy and are motivated to play.Accessibility: Video games can be accessed remotely, making the therapy more convenient for individuals who may have difficulty traveling to a therapy office.Assessment: Video games can be used to assess an individual’s cognitive abilities, such as memory, attention, and decision-making skills.Virtual Reality: Video games can be used in virtual reality environments, allowing individuals to experience immersive and realistic simulations of real-world scenarios. This can be particularly beneficial for individuals with phobias or other anxiety disorders.

All that said, there are different examples in the literature about how to take advantage of the use of video games to support (psychological) therapy (e.g., [6]). Still, few describe an integrated approach such as what we propose here as Video Game Therapy^®^ (VGT). In the following lines, we describe the genesis of the approach, its use, and the psychological methodologies and tools employed to support the therapy process. The primary aspect of VGT^®^ is the idea of having an integrated approach where the gaming experience of the patient is at the center of the therapy.

## 2. Video Game Therapy—Putting Video Games Together with
Therapy

Video Game Therapy^®^ was created in 2019 by Dr. Francesco Bocci, taking inspiration from Geek Therapy [7] and classic psychodrama [8]. It was designed as a tool to be used in psychology to allow for emotional containment, clinical and therapeutic work, as well as supportive and expressive work. The idea behind the tool is using commercial video games to reflect on several aspects of one’s lifestyle, emotions, and thoughts that come to life in the game setting. This approach is exceptionally versatile and can be used with patients of a wide age range, not just children or teenagers, but also young adults and adults. Video games offer the possibility to interact in an imaginary scenario, visually concretized through video–digital support. In this scenario, the patient can potentially express the most salient aspects of themselves with complete freedom and fewer defenses compared to exclusive dialogue, thanks to the immersive properties of video games that make the gaming experience particularly spontaneous. Despite the initial skepticism, as video games still struggle to be recognized as a potentially reparative tool in clinical settings, a positive response was recorded during and after the pandemic. In fact, while the pandemic has increased the social incidence of mental health disorders, it also substantially contributed to heightening the need for accessible and cost-effective coping methods, of which video games represent a viable and effective solution [9].

The scientific principles that support Video Game Therapy^®^ are based on the evidence provided by numerous studies investigating the use of video games as cognitive training, neuropsychological rehabilitation, forms of learning, and recovery of impaired functions. Contrary to common belief, skills developed within a video game framework are not confined to the virtual environment. Instead, they are likely transferred to individual and relational contexts of everyday life. That is why, according to numerous critical issues, commercially available video games and games developed specifically for therapeutic use have been previously employed for rehabilitation enhancement such as traumatic brain injury [10], neurodegenerative diseases, DSA [11], ADHD [12], as well as in the prevention of cognitive decline in physiological aging [13]. For instance, research has shown that intensive training with action video games can improve the recovery of the right hippocampus, a brain area crucial for spatial navigation and often compromised in neurodegenerative disorders and cognitive decline. Based on this, a research group at the University of Turin’s Department of Psychology has created “MindTheCity!”, a 3D video game improving the efficiency of brain circuits associated with navigation and spatial memory. Moreover, on 15 June 2020, the Food and Drug Administration (FDA) approved using a video game as the first digital medicine treatment. Endeavor rX is an action video game that can be prescribed as a treatment for ADHD in pediatric patients, as it trains attention abilities when performing two or more tasks in parallel. The video game was developed by Adam Gazzaley, a neurologist and professor at the University College of San Francisco, in collaboration with game designers from the industry. The approval process required years of clinical trials involving 600 children aged 8 to 12. The study empirically supports the idea that digital treatment can support pharmacological therapy in improving reading skills in children with DSA [14] while reducing therapy access problems for patients and families with transportation and mobility issues. Video games can also be suitable for children with ADHD diagnosis. One might expect children with ADHD to have difficulty concentrating on the video game, as they report difficulty maintaining attention over time. However, thanks to the engaging nature of video games, these tools are ideal for increasing focusing skills [12,15,16]. It has also been found that children with higher attention deficits achieve more remarkable attention improvement through computerized training and show a persistent effect in nine months [17], with players with lower basic reasoning abilities obtaining more significant cognitive benefits, such as better divided attention and faster stimulus perception [18]. Video game interventions have also improved social skills, communication, and motor function in children diagnosed with ASD [19]. Interestingly, the effectiveness of commercial video games in reducing anxiety levels has been demonstrated across various game genres (from exergames to action games) and gaming devices, including smartphones [20]. Furthermore, several studies have also suggested the effectiveness of various video games as a treatment for depression, reporting high adherence rates, acceptability, and feasibility [21]. For instance, a single 30-minute-long game session of Plants vs. Zombies significantly contributed to reducing treatment-resistant depression symptoms in adults [22]. Finally, regarding PTSD, Tetris-based intervention paired with psychotherapy has proven to be a successful method to reduce symptoms of PTSD and intrusive memories, as reflected in an increased hippocampal volume in both combat-related PTSD [23] and PTSD due to motor vehicle accidents [24]. However, benefits are not limited to patients with a psychiatric diagnosis. For instance, commercial video games such as Crash Bandicoot have been proven to be effective ways to improve task planning in children with difficulties respecting rules at home, and Little Big Planet effectively sustains conflict management through discussion, compromise, and adaptation techniques [25].

Video Game Therapy^®^ allows patients to express themselves in a protected and fun environment, and it also allows the therapist to understand the patient’s emotional state through their game choices and behaviors. The second foundation of VGT^®^ lies in the work of the psychologist Anthony Bean, founder of the innovative Geek Therapy [7,26]. This approach uses geek artifacts such as anime and manga, Marvel and DC comics, superheroes, video games, role-playing games, LARP (live-action role-playing), and pop culture for therapeutic purposes. Geek patients identify with the protagonists of their favorite artifact, whose fictional dimension allows for unrestricted expression of emotions, thoughts, and behaviors. In these experiences, the social judgment of the real world is suspended, and the context is therefore sufficiently protected for people to open up while having fun at the same time. In addition, this setting allows for observing the patient’s experience and understanding their feelings and experiences. This approach is suitable for all ages if patients have geek interests, and it has been found helpful in treating self-esteem problems, anxiety, depression, and post-traumatic stress disorder. It has also benefited patients with ADHD and autism spectrum disorders. Consequently, Video Game Therapy^®^ was created as a clinical therapeutic protocol by combining empirical evidence supporting the effectiveness of video games in clinical settings and Antony Bean’s Geek Therapy approach. Developed in Italy by one of the three authors [27,28], Video Game Therapy^®^ uses video games as tools for emotional regulation and personal growth in children, adolescents, and adults. It allows patients to express themselves in a protected and fun environment and allows the therapist to understand their emotional state through their game choices and behaviors. It is important to remember that Video Game Therapy^®^ should not be seen as a replacement for traditional therapy but as a complement.

Currently, Video Game Therapy^®^ has brought positive results in the following areas:Promoting emotional experimentation with the other (therapist), activating emphatic processes, through activating the emotional and affective sphere.Promoting emotional literacy;Encouraging awareness of the primary emotion of “discovery”, as compensation for an inferiority complex;Promoting a social feeling, the game becomes a tool for mutual cooperation and sharing of internal dynamics;Promoting awareness of one’s role or identity at a given moment in life, thanks to the correlation between the real-life lifestyle of the game and the protagonist’s avatar or a character in the virtual story;Activating cognitive processes of imagery, ideomotor training, and self-regulation;Reactivating the state of Flow, promoting concentration and mental processes related to attention, as well as problem solving, critical thinking, team building, proactivity, and decision making.Facilitating communication of parts of the Self.

The results have been primarily related to improving self-awareness and emotional states but also include reflections on how one’s skills are used during gaming, making correlations with real-life situations. It is fundamental to highlight that the focus of the process is not so much related to the medium (i.e., the video game) but to “how” the therapist presents and uses it during the session. Seeing therapeutic gaming as merely a tool to train skills, as a distraction and escape, or as occupational therapy would be very reductive. Instead, Video Game Therapy^®^ is a way to explore emotions and feelings in a protected environment, where the patient can experiment with different scenarios and characters, learning from their experiences and reflecting on how they can apply them to their real life. Such an imaginative setting allows patients to relive projections, identifications, emotional experiences, past traumas, and childhood memories. Then, the challenge for the therapist is to make these emerged contents find a sense of finality in the patient’s current life situation, offering new interpretative paths and new insights and, ultimately, using these contents to operate a process of transformation and therapy. Therefore, the focus lies on the actively playful relationship between the therapist and the patient. Playing is an imaginative space where people can become unproductive compared to what the world asks of them. This form of “presentification” (Flow) is often missing in our time and yet contributes to healthily exercising the Creative Self. As can be easily guessed, playing is not an entirely rational act. Therefore, to avoid the risk of favoring chaos in the creative process, the therapist should always guide the individual patient to find personal meaning in the contents elicited by therapeutic playing, mirroring digital stories with their real-life experiences and vice-versa.

## 3. Therapeutic Process in Video Game Therapy and Tools

As previously stated, Video Game Therapy^®^ puts video games and the gaming experience at the center of the therapy. It mainly concerns two characteristics of the gaming experience: Flow and relational setting. Flow is usually associated with the gaming experience. In contrast, the relational setting is related to therapy and refers to the alliance built between patient and therapist. On the one hand, flow experiences activate working, spatial, and visual memory while playing, balancing the game’s challenges and sustaining the player’s immersion in the gaming environment. On the other hand, the relational setting allows for a deep connection between the patient and the therapist, facilitating the flow dynamics created through gaming. Combining these two concepts enables the player to have a fruitful gaming experience alongside a therapist who helps them play and leads them through the game. From this perspective, VGT^®^ is not a standalone product but an approach that integrates psychological techniques and tools within a support and therapy pathway.

### 3.1. The Different Phases of the Process

VGT^®^ was elaborated in light of the Adlerian therapy vision. Adlerian therapy is focused on four subsequent stages. First is *engagement*, where the main goal is establishing trust between the therapist and the person in treatment. Second, there is the *assessment* phase, where the therapist invites the person to speak about their personal history, family history, and other aspects of their lives, such as overall lifestyle patterns. The third phase is called *insight*. Here, the therapist helps the person to create new ways of thinking about their present condition. During the last phase, *reorientation*, the therapist helps the patient engage in satisfying and practical actions that reinforce or facilitate the individuation of new insights. Figure 1 presents how the different phases of Adlerian therapy and VGT^®^ match and outlines the ways how the method unfolds. In the following sections, we will describe the different phases of the VGT^®^ approach in more detail.

### 3.2. Assessment and Play

Once established a trusting relationship with the patient, the therapist can devote themselves to assessment. The assessment phase in VGT^®^ is mainly about meeting with the patient to know them and delineating the therapy’s general goals. However, in this phase, it is also fundamental to identify the patient’s characteristics in terms of their personality. The definition of the patient’s personality is vital to help the therapist choose the best video game for the individual. Personality investigation is done using the Myers Briggs Type Indicator (MBTI) [29], which returns one of the sixteen possible typologies. Other instruments have been used in the literature to identify the player type, from the Big Five [30] to Bartle’s player typologies [31]. However, the MBTI’s definition of the four dimensions as dichotomous components describing how the person/player behaves towards the world makes it easier to pair their gaming style with the type of video game that best fits that personality. Moreover, MBTI can be associated with more game genres than Bartle’s player typologies, as the latter was based on online multiplayer games. Linking a particular game to a specific personality can increase the emotional and intellectual involvement of the player, making it easier to reach the flow experience and achieve transportation and transformation during the therapeutic process. However, selecting a suitable game is not only associated with personality type. Other intervening factors include the diagnosis, the therapeutic goals, the patient’s current life, and the risk factors to be avoided. Risk factors refer to games in terms of gameplay, for example, by avoiding anxiety-inducing video games (horror, timed games) and in terms of narratives that call for self-exploration if they are not part of the therapeutic intention. The authors created a set of video game description forms to help therapists choose the aptest videogame according to the patient’s situation, emphasizing different aspects relating the games to the patient. In these, video games are framed according to various criteria: PEGI, description, genre, psychological description, area of intervention, and MBTI typology.

Table 1 presents a list of commercial games associated with the patient’s MBTI typology.

### 3.3. Gaming and Flow State

Play is central to Video Game Therapy^®^, and patients experience its true power as the therapist aims to bring them into a flow state.

The term “Flow” [32] refers to a particular psychophysical state in which people are completely immersed in what they are doing, to the point of losing their sense of time and having exceptional perception levels. Their senses are amplified to the point of altering perception: the field of view seems larger, external noises disappear, smells become more intense, and the body feels lighter. There are no distractions, as attention is entirely focused on the stimuli relevant to performance. Consequently, people in a flow state do not feel tired, as the feeling of being able to continue their activity indefinitely prevails. Everything happens fluidly and harmoniously during this state, and all psychophysiological systems work together. Nonetheless, Flow is not just purely positive feelings, as it also carries the properties of effort and frustration, provided it is not excessive. These negative sensations lead to improved self-control during the task. For example, players of a horror video game (Resident Evil 7), despite registering a significant amount of anxiety due to the video game’s themes, also showed an increase in perceived happiness after a short gaming session [33], as reported by the Visual Analogue Scale for measuring Perceived Anxiety and Basic Emotions [34].

There is a wide range of terms related to the state of Flow. For example, the literature alternatively defines it as an “optimal experience state”, a “state of grace”, and a “zone of maximum performance”. Colloquially, people refer to feeling in a Flow state as “ being in the zone”. Despite differences in terminology and definitions, longing for this state seems to be transversal to different cultures, as confirmed by numerous trans-cultural studies highlighting how humans prefer to employ their psychic resources in environmental opportunities that offer the possibility of experiencing this complex and rewarding state of consciousness [35].

Csikszentmihalyi has identified the elements that characterize the state of Flow as follows:**Clear goals:** Subjects identify precise short-, medium-, and long-term goals and plan how to achieve them.**Total concentration on the task:** A high degree of engagement in a limited field of attention on the present. Attention is entirely focused on the action.**Loss of self-awareness:** The subject is so absorbed in the activity that they are unaware of it. They are aware of their actions, but it is as if they are not aware of this awareness.**Distortion of the sense of time:** Time perception is altered as one is completely absorbed in the experience and unaware of the passing of time.**Direct and unambiguous feedback:** The effect of the action is perceived by the subject immediately and clearly, providing unequivocal feedback.**Balance between challenge and skill:** The activity is proportionate to one’s abilities (therefore, there is no boredom or anxiety).**Sense of control:** A perception of complete control and ability to dominate the situation.**Intrinsic pleasure:** The action provides intrinsic satisfaction, which is an experience that is highly rewarding and satisfying, so much so that expressions such as “addicted to victory” or “addicted to success” are used.**Integration between action and awareness:** Concentration and effort encourage individuals to practice that activity, learn to understand their sensations better, and rediscover the maximum connection between mind and body.

The Flow model is defined as a middle ground, a funnel zone in which opportunities for action (challenges provided by the activity) and the individual’s potential (skills) are balanced. Outside of this zone, if there is an imbalance in favor of challenges, the individual may experience anxiety as they do not feel adequate for the task. On the other hand, if the imbalance leans towards skills, the subject may experience apathy and boredom, as the job requires little effort and is then perceived as un-stimulating. Games—and video games— can easily favor players’ entry into the flow state. Players are often so immersed in the game activity that they lose awareness of the time and space around them. According to Huizinga, players immerse themselves in a “magic circle” [2], a protected area that guarantees high levels of psychological safety and stimulation through fun and intrinsic motivation, desire to experiment, and curiosity. In fact, playing represents a rewarding and satisfying experience in which the boundaries between the space of action and intention and those of time become increasingly blurred. The result is a deep concentration in which the fear of failure gives way to the joy and pleasure of doing [36]. It must be underlined that the state of Flow is more significant in video games than cinema, for instance. Flow is not achieved solely through mere immersion but requires the above-mentioned balance between challenges and skills. In cinema, the flow experience is minimal because although the viewer is immersed in a story, no active skill is required except for understanding the story. By contrast, in video games, the balance between challenge and skills is achieved through gameplay, as players act over the game’s world, controlling game elements and their avatars. Flow can be achieved in a simple arcade game by balancing the game’s demands and the player’s working memory and reaction times. On the other hand, in narrative games, the player controls the story’s direction and the dialogues, slowly immersing themselves into the story and identifying themselves with the characters. In this case, a state of Flow is achieved more gradually but as intensely as in gameplay-based genres. By directing the player’s attention to both their skills and emotional states, video games represent an optimal tool to activate insight. While in this state, players can safely disassociate while still managing the dimension of self-control while the game provides clear, defined goals and precise feedback. This unique state represents an immersive and transformative experience. It leads the player to be less defensive and more open to self-regulation, reaching a condition similar to what occurs with MDR. However, in this controlled hypnotic technique, a person talks about something while focusing on something else. Nonetheless, in the case of Flow, the person is focused on something they have agency over, not something that simply occupies their mind. It has to be underlined that, even in games, there are situations in which a state of Flow cannot be reached. It happens when the video game asks too much of the player and frustration ensues, or, by contrast, the game bores the player by asking asks too little, creating a negative compensation, as Adler would say, to an existential void. In conclusion, reaching a state of Flow depends on the challenges provided by the game. On the other hand, it also depends on the player’s characteristics. In addition to constantly improving their skills, players must also be capable of maintaining self-control, despite being immersed in deep emotions. Moreover, some personality traits appear to be related to an increased possibility of experiencing Flow (e.g., people with a high degree of Conscientiousness in the Big Five Questionnaire [37]).

### 3.4. Insight Phase

During the insight phase of Adlerian therapy, the person in treatment is helped to develop new ways of looking back at how past experiences might have shaped their current beliefs and behaviors and initiate new images about their future. The therapist might collaborate with the patient by offering an interpretation; however, it is up to the patient to decide whether the suggested theories are accurate and useful. In VGT^®^, the insight phase consists of complete immersion into a compelling video game story and gameplay, allowing the patient to safely explore their recurring negative emotions, such as frustration, fear, or sadness, while in a protected and safe space [38]. In fact, all games (including video games) can be defined as “half real” as they elicit real emotions in fake environments [39]. In this phase, Emotional recognition and containment are of vital importance. First, the patient trains their emotional intelligence by recognizing their internal states and emotions [40]. Then, the second step is accepting the negative emotions and the events that caused them. During this process, the therapist uses their guidance to contain the patient’s energy. However, video games themselves can represent valuable tools in sustaining the patient’s emotional management [41]. For instance, according to a recent study conducted in Italy [41], the survival horror video game Slenderman: The Eight Pages can improve emotional intelligence. One hundred and twenty-one adolescents were randomly divided into two groups. The experimental group played Slenderman for one and a half hours weekly for eight continuous weeks, while the control group did not play video games. Asked to answer the Italian Version of the Emotional Intelligence Scale [42] before and after the experimental period, the experimental group improved compared to the control group in evaluating and expressing emotions in relation to the Self. Most interestingly, the beneficial effect lasted up to three months after the end of the experimental trial. Just as a caregiver would collect the child’s emotional despair and replace it with something comforting, video games will respond to those needs, redirecting them to new functional beliefs and behaviors. In this context, video games act as a compensatory technique, encouraging patients to safely explore new emotional management strategies that can be generalized to real-world situations.

Numerous commercial video games are increasingly often portraying stories with a profound emotional impact, sustaining self-reflection in players, allowing them to “peer into the dark reaches of the very real human heart” [43] while eliciting and controlling specific unpleasant emotions [44]. Examples of dark-themed video games include playing the role of a terrorist in Call of Duty: Modern Warfare 2 or using chemical weapons in Spec Ops: The Line. However, according to research, improvements in emotional management are not limited to adult-themed video games. Various video games, from Mario Kart to Doom, have been reported to help manage stress and support adaptive coping, eudaimonic well-being, and socialization skills [45]. For instance, a study explored the effects of a puzzle game (Sushi Cat 2) on players’ emotional states [46]. After a 6-minute session, the players showed a more positive emotional state and a lower stress level than those of another group of young adults who were instructed to engage in a 6-minute-long mindfulness session. A similar study examined a 20-minute-long gameplay session of a casual game (Flower). The game session led to a statistically significant reduced perceived stress in young adults. Still, in this case, the results were not as satisfactory as those obtained after an equally long mindfulness meditation session [47]. The insight phase encompasses different recurring themes, which will be further examined in the following sections.

#### 3.4.1. Knowledge of Feelings of Inferiority

Feelings of inferiority can be found in a patient’s history and might emerge during a gaming session in the insight phase. As previously stated, these episodes should be addressed by finding compensatory emotions through the game. However, during video game playing, the patient’s awareness of feelings of inferiority might be mitigated as a natural relationship exists between playing and failure. Video games open a new scenario on how failure could be made more acceptable through the lens of the game, leading patients to talk about these experiences differently through the game than in real-life contexts.

In today’s competitive society, both socially and economically focused on achievement and success, the possibility of failure is often minimized or denied [48]. The motivation to compete, therefore, often represents an attempt to distance oneself from the risk of failure, and the consequent avoidance of guilt, rather than a search for pride in victory [49]. What happens during gameplay differs, giving rise to *The Paradox of Failure*. Even though we tend to avoid failure, we look forward to playing, despite this activity usually making us experience something we would naturally want to stay away from [50]. This is because the game itself, if designed well, pushes the player to “try again and fail, but fail better” as quoted by Beckett [51]. This approach encourages players to freely explore possible solutions in the safety of a realistic yet harmless context.

In a video game, failure can be defined as “the inability to make progress towards the goal” [52]. Yet, we can identify a taxonomy of different fiascos under the umbrella term of “failure” [52]. For example, in chess, we expect most challenges to revolve around the failure to form effective plans. It is usual for a player to fail because they created an ineffective action plan. However, if players fail consistently because the chess pieces are hard to grasp and keep slipping, then this leads to a different type of failure, as moving the chess pieces should not be the main challenge of the game [52]. Keeping this example in mind, we can define two macrotypes of failures: *out-of-loop* failures and *in-loop* failures [52].

Out-of-loop failures represent unintentional and unexpected failures. Specifically, these are the actions for which players do not even anticipate the potential for failure, which results in players (and sometimes the therapist) being shocked if the failure occurs [53]. Therefore, this type of failure should be minimized as it negatively affects the game experience. Related to out-of-loop failures is game-related frustration: in the context of VGT^®^, it results from mistakes due to misunderstandings between the therapist and players, unclear rules, and anything that delays or confuses the players’ actions during the session, detracting from the sense of immersion [54]. It should, therefore, always be avoided.

On the contrary, we can designate the failures contemplated by the game’s design intent and its rules as in-loop failures. This type of failure is not harmful to the enjoyment and engagement in the game. On the contrary, this failure category is often the primary component that makes the game challenging and engaging [52].

In-loop failures include actions with high uncertainty, leading the player to expect and contemplate the possibility of failure even before taking action [53]. Linked to in-loop failures is in-game frustration, which must be preserved per the “trial-and-error” nature of games. This “productive failure” facilitates understanding the issue in question, encouraging the design of its solution [55,56].

In such a context, players must be clearly informed of their progress, identifying the degree of discrepancy between their performance and the goal of the action [57]. Failure and frustration become a way for the player to contemplate and reformulate what it means to him to be in a difficult situation. Therefore, these moments have an “interesting tension: […] leading people to think about why they don’t stop playing” [58].

It is, therefore, important to emphasize that losing in a game does not always mean defeat [59]. Failure can be correctable, representing the individual’s growth and the perfect tool to confront difficulties constructively by learning from the past. This is consistent with one of the rules of Flow, which is the perfect balance between the player’s abilities and the game’s requests. Proposing tasks that are easy enough to master but challenging enough to stimulate the player’s attention and desire for success, games support the player’s motivation through a push to success, “challenging them optimally” [36]. Playing is thus a process of continuous player growth until a sense of security and control over what is happening in the surrounding environment is reached [32]. In-game frustration is, therefore, temporary and mediated by the certainty that one day the player’s skills will be enough to face increasingly hostile enemies and increasingly tricky situations. According to Self-Determination Theory, a fundamental component of an individual’s determination to complete a given task is the sense of competence, the feeling of being competent in what one is doing [36]. Players perceive competence when performing actions that prove to be effective or completing the proposed objectives [60,61].

In particular, in Role-Playing Games, one can talk about retrospective meritocracy, which defines merit in terms of achievements in the past; Role-Playing Games reward the player based on what they have already achieved, not on their potential [62]. This is a crucial factor if applied to everyday life: the retrospective model ignores the social context and focuses on real merits, based on actual events [63]. In conclusion, by managing to generalize the sense of control beyond the gaming environment, an increase in the player’s confidence might reflect an increase in self-esteem and motivation for the task [64].

#### 3.4.2. Catharsis and Video Games

The expression of experiences of inferiority can also be addressed through catharsis in video games. Catharsis is a term in dramatic art from Aristotle’s conception, that describes the “emotional cleansing” the characters or the audience often go through while watching a play. It represents an extreme change in emotion resulting from experiencing strong feelings (such as sorrow, fear, pity, or even laughter). In modern times, catharsis is defined as an emotional release eliciting feelings of restoration, renewal, and revitalization. According to psychoanalytic theory, this emotional release is linked to a need to release unconscious conflicts and is very similar to sublimation. For example, rather than vent work-related frustration, an individual may release such tension through physical activity or another stress-relieving activity. Another approach to understanding catharsis in video games draws from Moreno’s psychodrama principles, emphasizing the therapeutic value of acting out emotions and experiences in a controlled setting.

The psychological sphere considers sublimation as releasing an unacceptable social impulse through a different tolerable activity, including painting, reading, or playing video games. Not coincidentally, most video games have murder or violence as the basis of their gameplay. The ability to perform a fantasy action that would result in concrete consequences without repercussions is an outlet for the individual, who can both sublimate a harmful emotional state and “purify” themselves from it.

However, the idea of catharsis through violent video games is a highly debated topic. Some experts argue that violent video games can allow players to release their aggressive impulses and negative emotions in a safe and controlled environment. As studies have reported a negative correlation between violent game use and violent crime performed in real-life situations, the relationship between video game aggressiveness and real-life aggressiveness might be “closer to displacement rather than imitation” [65]. Others argue that violent video games can lead to desensitization to violence and have a negative impact on players’ behavior (e.g., as discussed in [66]).

Although it is questionable whether catharsis can be achieved through non-violent video games, many games aim to evoke emotional responses in players without relying on violence, such as puzzle games or story-driven games. However, this is a delicate issue. Therefore, it is essential to recognize that the impact of violent video games on individuals can vary greatly, depending on their personal experiences and emotions.

Ultimately, the effectiveness of video games as a form of catharsis will depend on the individual player and their specific needs and experiences.

#### 3.4.3. Desensitization

Cognitive Behavioral Therapy and exposure therapy are among the most common interventions in treating anxiety disorders, traumas, and phobias. Exposure therapy stems from classical conditioning and is based on habituation, in which patients maintain gradual, repeated, and prolonged contact with fearful stimuli until the associated anxiety decreases [67]. In other words, gradually exposing the patient to the threatening stimulus helps them progressively “familiarize” with it, increasing their perceived safety and ability to control their negative emotions. The second step is information processing, meaning re-evaluating old information and incorporating new, alternative, and more functional information into the trauma memory [68,69]. For instance, several recent commercial video games have portrayed the psychodynamics of bonding, separation, grief, and even pregnancy loss [70]. Depending on the patient’s state, different intensities of exposure therapy can be advised. Systematic desensitization is the most commonly used method. It consists of relaxation training followed by gradual, repeated, and prolonged contact with the stimulus, starting from the least feared and building their way up to increasingly aversive images. The exposure is usually paired with precise practice tasks specified by the therapist, encouraging the patient to divert their attention to task-oriented matters. Systematic desensitization is based on reciprocal inhibition, proposing that two opposite emotions can not co-exist [71]. On the other side of the spectrum, “flooding” is a desensitization technique involving intense exposure to a challenging stimulus until the patient’s emotions towards it become neutral [72].

#### 3.4.4. Exposition to Stimuli

Traditional therapeutic protocols offer two main exposure methods. “In imagination” (or “imaginal”) exposure involves thinking about the aversive stimuli, allowing patients to confront and contemplate their unpleasant consequences in the form of a vivid story told by the therapist [73]. By being exposed to this narrative repeatedly, the patient imagines the scenario intensely, learning that dwelling on the worst possible outcome of their fear does not make it occur [74]. The slowly developed new perspective on the stimuli ultimately leads to a more objective appraisal of the antecedents of a perceived incoming catastrophe [75]. Imaginal exposure is deemed appropriate for specific cases, such as those that imply a lengthy chain of events resulting in a distal disaster or an extreme personality change (e.g., becoming a pedophile; [73]). However, in some cases, imaginal exposure may not be the best route, as the patient may trace back the absence of negative consequences to the unrealness of the situation. The second traditional method is “in vivo” or “real-world” exposure. It implies gradual and repeated contact with the phobic stimuli in a controlled and systematic setting. Contrary to what may happen with imaginal exposure, in vivo exposure produces disconfirmation, as direct contact with the stimuli does not materialize the associated disasters. Moreover, reincorporating the avoided activities into the patient’s daily routine seems to have long-lasting positive effects at follow-up [76]. However, despite its relevant advantages, in vivo exposure does involve several limitations. According to research, up to 25% of patients refuse or abandon treatment after considering the direct experience “too threatening” [77]. Moreover, the therapist’s office may not be adequately equipped for in vivo exposure, failing to guarantee the safety of the environment in which it occurs. Finally, the therapist might not be able to fully control several fundamental variables of exposure, such as the order in which the stimulus appears or its intensity. Technologically enhanced methods can, therefore, represent a viable solution in some cases. “In virtuo” or “virtual exposure” involves contact with realistic problematic stimuli in virtual environments, including video games and more immersive techniques, such as Virtual or Augmented Reality. In virtuo exposure therapy generally follows a standard path. First, the user is exposed to a virtual reproduction of a growing hierarchy of anxiety-inducing situations. Then, thanks to the therapist’s guidance, the patient learns to know and manage their dysfunctional responses through breathing and relaxation techniques until their anxiety level gradually decreases. Of course, it is possible to return to a less stressful level of treatment at any time or get out of the virtual world by removing the VR headset or turning off the video game. In virtuo exposure can have several advantages over traditional exposure techniques. On the one hand, it has the potential to be more immersive than imaginal exposure. Immersion is defined as “the sensation of being surrounded by a completely other reality […] that takes over all of our attention, our whole perceptual apparatus” [78]. Immersion can be divided into three sub-components: sensory immersion (involving sight and hearing), immersion based on challenge (which concerns the objectives proposed by the therapist), and, finally, imaginative immersion (involving patients’ emotions) [79]. Video games are known “for their ability to deeply immerse users, stimulate the senses, and tap into a broad range of emotions” [80]. Therefore, playing video games makes it easier for patients to maintain undivided attention for extended periods [81], facilitating access to different states of consciousness and helping them regress to childhood play [82]. When total immersion into the activity is achieved, the patient reaches the so-called sense of presence, the feeling of being physically within the simulated reality, seeing through the eyes of their avatar and experiencing real emotions [54].

Patients can, therefore, experience the novelty and challenge of fictional activities without real-life consequences [83]. It is also worth underlining that in virtuo methods may be perceived as more acceptable, at least to some consumers, and more enjoyable than traditional methods [84]. This is relevant, as engagement has previously positively affected compliance rates [85,86]. Moreover, in virtuo exposure ensures superior control of the fundamental variables of the phobic stimulus if compared to in vivo solutions. These include the number of stimuli, their movement, size, distance, order of appearance, and the ability to stop, resume, and repeat scenarios for as long as necessary, allowing an intervention tailored to the patient’s needs. This way, patients can experience a successful performance in a highly controlled environment. Bandura in [87] argued that repeated successes could increase the sense of self-efficacy, diminishing the impact of failures. In virtuo exposure, simulating highly complex, engaging, realistic, safe scenarios allows patients to build the necessary mental and psychophysical skills before engaging the phobic stimuli in a natural environment. Technology can thus become an “experiential interface” [88] in which knowledge is produced by embodied experience. Literature has repeatedly suggested evidence of the successes of in virtuo exposure therapy. For instance, Virtual Reality is a gold standard for treating social phobia and fear of public speaking. The proposed protocol allows the patient to experiment in front of an audience of virtual characters within situations recreated ad hoc, such as business meetings or dinners [89]. Positive results also include the treatment of the fear of flying, with a significant amortization of the time and costs of the intervention compared to in vivo exposure [90]. In addition, VR exposure has shown positive results relating to the treatment of arachnophobia [91], the phobia of driving [92], and acrophobia [93].

According to the literature, even non-immersive video games represent compelling in virtuo exposure methods. Although serious games have been developed to tackle problems such as eating and impulse control disorders [94], commercial entertainment video games have also been examined as a viable alternative or additional form of treatment [95]. Specifically, commercial video games present a unique benefit compared to more serious-minded in virtuo settings: they represent an effective tool for establishing a closer patient–therapist relationship based on shared interest, especially in cases where patients are young adults [96]. For this reason, commercial video games have been examined as viable in virtuo exposure tools over the last two decades. For instance, the commercial video game Full Spectrum Warrior has been previously used to clinically treat soldiers for post-traumatic stress disorder (PTSD) caused by military service [97], thanks to the extreme realism of in-game soldiers’ movements and behavior. According to another study, several commercial driving games (London Racer; Midtown Madness; Rally Championship) had a positive role in treating driving phobias developed after involvement in car accidents, provided patients showed sufficient immersion in the gaming activity. Furthermore, these results were confirmed even when co-morbid conditions, such as PTSD and depression symptoms, were experienced by the patients [92]. A different study compared the effects of a neurofeedback serious game (Mindlight, GEMHLab, 2016) with those of a commercial video game showing fearful obstacles (Max and the Magic Marker) on anxious children [98]. Although the commercial video game did not explicitly include evidence-based anxiety-reduction techniques, analyses revealed a significant anxiety reduction, as reported by the children and their parents, to the point that the magnitude of improvements was equal to the one reached after neurofeedback sessions. Finally, commercial video games such as Tetris have also been proposed as a protective mechanism against nightmares connected to traumatic events in military personnel [99] and a practical visuospatial task interfering with the recursion of sensory imagery from traumatic events [100].

### 3.5. Reorientation

The fourth and last phase of Adlerian therapy is called reorientation. At this stage, the patient is ready for change. Therefore, the therapist helps them engage in satisfying and effective actions they can use in their daily life, outside of therapy, to reinforce or facilitate the new insights gained in the previous phase. It is important to note that, in this phase, the therapist’s role is to accompany and amplify the patient’s sensations and imagination elicited by the selected video game’s narrative contents. Video game players often engage in spontaneous utterances, which can be used as a source of information regarding the internal states and reasoning of the patients themselves regarding story elements and characters [101]. In VGT^®^, the therapist listens to the patient’s point of view and encourages such exploration of novel ideas.

Starting points of reflection often arise from the patient’s identification with in-game characters. In fact, video games act as “subjectivation machines” [102] as they encourage the patient to simulate, adopt or temporarily experiment with different identities and points of view, thus preparing them to apply similar techniques in the real world [103]. The fourth phase, therefore, involves a transformation of the patient’s outlook on life and the creation of new effective strategies [104]. The following section, dedicated to storytelling and imaginative techniques, will further investigate the topic of reorientation.

#### Storytelling and Imaginative Techniques

During a VGT^®^ session, the therapist should listen to the player and play with them freely. During gameplay, and especially in a Flow state, players tend to communicate their experiences, worldviews, and deep needs through the games’ images, stories, and characters (avatars). In particular, an avatar allows players to explore their inner thoughts while manifesting them for the therapist to see. That is why avatars can be considered an interface of communication between the patient and the therapist and themselves. Furthermore, avatar identification can lead the patient to experience fascinating combinations of syntonic or dystonic emotional experiences between themselves and the character they portray, opening new sense-making opportunities [65]. The therapist should tune into the patient’s emotional experiences and system of representations, but it is equally important to listen when players speak through their in-game actions. For instance, a case study reported a teenager referred to psychotherapy for learning disabilities and relationship problems [105]. Interestingly, his style of play was heavily characterized by control and insistence on hygiene-related themes, revealing much more about the boy’s inner thoughts than initially reported. For this reason, in-game actions represent an essential compass toward the patient’s symbolization of their lives into a storytelling approach [105]. The therapist can more effectively recognize such a creative and childlike narration by temporarily abandoning adult mental structures and rationality and rediscovering the traces of childish thinking still present within them, radicated on an emotional and fantastic level. Through this, the therapist can successfully accompany the creative, childlike part of the patient they meet in the therapy rooms through Video Game Therapy^®^. Equally important is listening to patients, tuning into their emotional experiences and system of representations, immersing themselves in their way of seeing and experiencing the world, and embracing their expressive and communicative register. This is the only way the therapist can truly understand (in Latin, cum-prendere, meaning “to take together”) what players are trying to tell. That is why therapists should combine academic studies, field experiences, and in-depth work on themselves. The story of Luca, a young patient treated with VGT^®^, is an excellent example of the power of free storytelling. Luca suffered repeated abuse from his father and, even after about three years since his father’s removal, still lives in anguish caused by those traumatic events. Yet, the child in therapy does not explicitly talk about those events because he has already painfully recounted them several times during the legal trial in the previous years. The stories he tells through video game playing speak of his anxious experiences, particularly of a central aspect of his psychological trauma: a father suddenly becoming cruel. While playing Minecraft, Luca speaks of a King who enjoys torturing without reason. In other video games, Luca tells stories about characters possessed by demons, men who mask themselves to become evil, ambiguous figures that unpredictably become enemies. Even though these villains are imprisoned, eliminated, or killed, they return shortly after as persecutors in a story that repeats itself relentlessly. Luca did not directly disclose what his father did to him, but he often spoke about his distress through video game stories during therapy sessions. Over time, the child’s narratives have contributed to displacing the negative and aggressive feelings first associated with avatars and characters into actual people, such as family members, sustaining the patient’s sense-making of the present situation [65]. He talked about his disbelief and terror in front of a parent who inexplicably transforms, his disorientation in living with such devastating ambiguity, and a traumatic core that seems insurmountable because every time he tries to distance himself from it, it returns unchanged. While in a state of Flow, he shared his experience, and the therapist welcomed it through free listening, which left deep marks on the child’s representations of himself and the world. This included affective ambivalence, emotional discontinuity and incoherence in self-perception and perception of others, fear of aggression, and a marked distrust in relationships. This level of communication between the child and the psychotherapist is rich in meaning, authentic, and profound. It is a level on which psychotherapeutic work can begin to reassure, contain, rehabilitate, reinforce, redefine, and reorient. It allows for a representation and sharing of an experience that would otherwise remain mentally unrepresentable, therefore pure anguish, decidedly more devastating and uncontrollable, like all things without form and name. Through this level of communication, the inner world of patients can be expressed through scenarios, characters, and stories, as well as gameplay sections closely linked to the activation of skills such as attention and problem-solving. They represent the patient’s experiences, needs, anxieties, desires, and emotions, which can now be narrated, played, and thus shared with someone who knows how to listen. Through these forms of communication, Luca talks about himself much more than he does by answering direct questions. However, although the characters and settings of stories can represent the contents of the inner world, this does not mean there is a rigid and univocal correspondence between individual characters and specific mental contents. The video game medium opens up the use of imagination and creativity, whose importance in promoting psychological growth can be found in the practice of child psychologists and psychotherapists [106,107]. However, storytelling techniques prove to be effective in adults, as well. Resembling works in narrative psychology, emotionally impacting video game stories contribute to the patient’s self-reflection by being incorporated into their own life stories [108]. The patients organize such stories into a complex autobiographical narrative centered on their perceived identity [109,110] that they try to superimpose on reality. In turn, the therapist uses this material as a projective test, immersing themselves in the patient’s world and helping them rework and reorganize its contents. All of this can potentially change how the patient sees and experiences themselves and the world and allow them to face developmental tasks more adaptive and harmoniously. Working with imagination is not just about working with fantasy but may have an impact on the perception of oneself and the world and how one acts and behaves in reality. In fact, such narratives may contribute to forming their perceived identity, as how individuals build their story reflects their explicit and implicit ways of making sense of their real-world experience [111,112]. The dramatization that develops during the Video Game Therapy^®^ session is often a significant moment of evolution, especially in many cases of psychosis [113]. Playful dramatization and storytelling in the form of a narrative start from the body and progresses towards the symbolic and verbal. Working with imagination may allow the patient to express their inner world, to represent it, and, together with the therapist, to rework its contents, reorganize it, find solutions to present conflicts, and provide new possibilities for emotional evolution. Video game stories can be about the client, their inner world, and experiences of relationships with the outside world. This happens through a process of externalization. The gamer externalizes what is happening in the patients’ minds, eventually distancing themselves from the content of their inner world and seeing it as something external to themselves. In this way, they may better understand and organize their internal world. The narrative of video games, especially in interactive adventure games with multiple dialogue choices, proposes an evolution for such internal dynamics. Thus, it not only allows for their representation but also suggests future, more effective directions. Still, patients’ narratives can also pertain to the therapist. For example, patients can process what is happening in the therapeutic relationship by projecting different roles onto the therapist, such as father, mother, brother, guardian, executioner, victim, hero, or antihero. It represents a unique opportunity, as the therapist is someone with whom the patient can confront and dialogue. By constructing a shared story, the therapist can help them orient themselves in their inner world, contain it, differentiate it, redefine it, and restructure it so they can act successfully in the external world. The video game scenario becomes a “fairy tale” in which both the child and the therapist play roles, dramatize situations, and invent imaginary scenarios. The ability to think and speak about experiences—and therefore the ability to tell and retell them—combined with the ability to organize internal and external experiences into a coherent and continuous spatio-temporal organization constitutes an endpoint in treating more disturbed or younger patients. In contrast, with more advanced patients, it can represent the main tool already in the early stages of treatment [114]. Imagination is expressed and played in different ways depending on the gamer’s age, level of affective and cognitive development, psychological functioning, and possibly the type and severity of the disorder. We can consider video games as a container that can be filled differently and variably by each patient, according to their particular emotional needs at that moment. In sandbox video games (such as Minecraft and Animal Crossing), the narration is free and created by the patient themselves. The concept of unsaturation applies here, meaning that the gamer can “fill” any scenario or character they introduce with different meanings at different times in their growth or under the influence of different needs and emotional states (or parts of themselves). This perspective refers to the psychotherapeutic model of Internal Family System^®^. Following more traditional psychoanalytic models [115], the therapist works with the material brought by the patient in the session, producing interpretive hypotheses that are nothing more than the decoding and explicitation of underlying and represented mental contents. The characters and scenes brought to the session are exclusively internal objects of the patient projected onto the therapist, who acts as an interpreter for these projections based on a code constituted by the reference theory. An alternative proposal to decoding meaning is instead the construction, through storytelling techniques, of a shared sense that leads to a common understandability [116]. This occurs through the therapist’s acceptance of the experiences of the patient. Their subsequent actions are configured as a form of containment, transformation, and restitution to the child in a form they can assimilate and tolerate. This process is similar to literacy; however, the therapist aims for an unsaturated sense, leaving room for any modification the patient may want to make. The VGT^®^ approach lies in seeing patients as potential creators and not just consumers. The goal is to help them overcome creative impotence, which is the inability to imprint their own mark (of desire, direction, strength) on the feelings and actions of everyday life. To do this, a therapist who enjoys playing video games and provides the patient with the space and time to do so as the protagonist is needed.

## 4. The Foundation of the Individual Psychoanalytic Therapy Applied
to Video Game Therapy

In this section, we reflect on what makes VGT^®^ peculiar and not simply another way to use commercial video games in therapy. This reflection is rooted in considering VGT^®^ as an elaboration from the foundations of the Individual Psychoanalytic Therapy proposed by Adler [117]. Through the process of empathetic encouragement based on containment, the Adlerian therapist tries to share the impenetrable personal logic of the patient. In turn, this affects the therapist’s own logic in a mutual game of “contamination”. Finally, a shared “common logic” is generated from this reciprocal interaction, defining the establishment of a creative couple [118]. This creative couple is at the core of VGT^®^.

We take inspiration from Winnicott’s view regarding the role of play in psychotherapy: “Psychotherapy takes place in the overlap of two areas of playing, that of the patient and that of the therapist. Psychotherapy has to do with two people playing together. The corollary of this is that where playing is not possible then the work done by the therapist is directed towards bringing the patient from a state of not being able to play into a state of being able to play […]. This gives us an indication for the therapeutic procedure: providing the opportunity for a formless experience and creative, motor and sensory impulses, which are the substance of play. On the basis of play, the entire existence of man is built as an experience” [119]. Therefore, this expression of psychotherapy as a form of play between the patient and the therapist is an instance of the creative couple’s possible manifestations.

The therapist covers two main functions while playing in a psychotherapeutic setting. We use the term “function” in the sense of Propp, referring to the role and actions specific to a particular character within a story, without which it is impossible to predict the expected changes in the story [120].

First, the therapist must ensure that psychotherapy is configured as a play space following the characteristics of playing, including its intrinsic pleasures, spontaneity, freedom, and uncertainty [2]. This play space is disconnected from real-world anxiety that might interfere with the patient’s thinking processes. It is, therefore, often defined as “transitional”, as it lies between the safety of virtual life and the impacting themes of the real one. Most importantly, this safe place initially belongs to the inner world of the patients, and meeting there allows the therapist to connect with them [65]. Since free play must be protected from interferences that the patients themselves could produce, it is necessary not to consider the play situation as the psychotherapeutic encounter’s starting point but rather a goal that must be gradually reached with conscious actions.

Consequently, the therapist is the guarantor of flow and total immersion in the activity. This responsibility starts with choosing the suitable game for the patient, according to their lifestyle, personality, condition, and other forms of expression. However, as therapy involves a creative couple and not an individual, this function does not end with training the patient to reach the flow state. The therapist also puts themselves in the position to reach a flow state with the patient so that such “networked flow” [121] may amplify its therapeutic effects. Moreover, the therapist leads the patient to take advantage of flow’s potentialities, encouraging free exploration. Yet, the therapist stays close to the patient, providing them a sense of safety while preventing harm in moments of possible frailty.

Therefore, the therapist’s second responsibility is being available for the patient during play whenever they become more fragile and open to communicating about themselves and their trauma. That is why “the therapist must have a mind for two, energy for two, hope for two, imagination for two” [122], especially when the space of play risks collapsing under feelings of anxiety and uncertainty [123]. In fact, the therapeutic relationship serves as an “emotional corrective experience” allowing patients to write and rewrite their stories and dissolve the knots and blocks that impede growth.

In short, the therapist’s responsibilities include creating and maintaining a good working alliance through an emotion-based relationship setting, as emotional change is often deemed necessary for long-lasting change in the patient’s well-being [124].

As previously stated, the insight phase of Adlerian therapy consists of reflecting on the patient’s past experiences and how they might have shaped their current beliefs and behaviors. According to research, emotion is a fundamental source of information regarding their environment, personal goals, concerns, or needs [125,126]. Besides concerning external events, emotions are also vital tools of self-construction and a crucial determinant of self-organization. In short, emotions reveal how individuals appraise themselves and the outer world, giving “life much of its meaning” [127]. In this context, it is the responsibility of the therapist to sustain the patient’s ability to use emotions functionally so that reminders of past traumatic experiences are not seen as a return of the trauma itself. Of course, however, maladaptive emotional interpretations of external events are challenging to change [128]. Therefore, the challenge of effective psychotherapy is collaborating with the patient to transform them into satisfying and practical actions, i.e., reorienting them. One of the most recognized evidence-based treatment approaches, ranging from depression to trauma and interpersonal problems [129], is Emotion-Focused Therapy (EFT). It was developed by Greenberg and colleagues in the 1980s following several empirical studies on the change process (e.g., [130]).

Within EFT, therapists act as Emotion Coaches, helping patients become aware of, regulate, and transform their emotional experience. It is crucial to notice that “coaching” implies the patient actively participating in the process, forming an empathic, highly collaborative, and validating relationship with the therapist, where both sides are held mutually accountable [127]. Such a relationship is, therefore, strictly “tailored” or “person-centered” [131]. This means the therapist is encouraged to enter the patient’s internal frame of reference and empathically follow their experience, adapting their intervention to facilitate their specific affective or cognitive processing issues [129]. In turn, this relationship facilitates the creation of a peculiar relational environment characterized by safety and support during the patient’s internalization of new ways of processing experiential information [132]. This setting helps reduce relational anxiety by providing interpersonal validation, allowing patients to regain access to their internal resources.

Through creating and maintaining a successful alliance involving relationship and setting, the therapist can effectively guide the patient’s attention toward their emotional processes, first and foremost, giving a name to otherwise unformulated personal experiences [127]. After identification comes acceptance, and patients need to be coached to welcome their previously constricted emotional experience. Then, patients learn to regulate unhealthy emotions thanks to the safe, calming, validating, and empathic environment created by the creative couple. This is crucial in assimilating past events into their ongoing self-narratives [129]. At his stage of therapy, the relationship of reciprocal trust between therapist and patient becomes even more crucial as the reorientation phase revolves around the topics of following and leading, operating on a recurring structure.

The therapist must resort to experiential methods as patients cannot be verbally taught new strategies for dealing with difficult emotions. The therapist begins by following the patient’s exploration of internal states, asking them to recount the emotional processing they are engaging in. By confirming their availability as open listeners, therapists provide a much-needed feeling of acceptance and validation of the patient’s inner states.

Then, the therapist can lead the patient towards novel emotions by directing their attention toward previously overlooked aspects of the situation. Focus shifting can be carried out through various techniques, including enactment [133]. Social psychology has famously highlighted the impact of role-playing on attitude change, as performing actions while playing a role allows people to experience beliefs in line with the position [134]. However, VGT^®^ can represent a more efficient technique than enactment based on imagery and artistic performance, although maintaining similarities with improvisation theater [135]. As previously stated, patients’ identification with in-game characters encourages them to explore different identities and points of view [103], leading them to experiment with novel emotions. Once made apparent, these feelings can transform the patient’s previous perceptions of themselves, their surrounding environment, and their reasoning on past events.

In conclusion, Video Game Therapy^®^ represents not only a possible application of therapy through commercial video games but also an approach rooted in Flow experiences and in the unique role of the therapist in the creative couple as a fundamental characteristic of Individual Psychoanalytic Therapy.

## 5. Conclusions

This article aimed to present the integrated approach of VGT^®^ organically and systematically. VGT^®^ was defined in 2019 by one of the authors [28], taking inspiration from Geek Therapy [7] and Moreno’s Psychodrama [8,136]. However, the literature reveals diversity in practices, causing confusion on the definition of setting, methodology, and the therapeutic couple’s dynamic role. Furthermore, according to several authors (e.g., [137]), there is also a need for systematizing the match between the patient’s diagnosis and needs and the material characteristics of the video game (i.e., mechanics, dynamics, and storytelling elements). Therefore, this article emphasizes the peculiarities of VGT^®^’s integrated approach compared to other methods that use video games as a therapeutic tool. In particular, the peculiarities of this approach originate from Adler’s Individual Psychology, where the therapeutic relationship is at the center, and the therapist becomes the guarantor of the flow. It is important to address the fact that the use of video games in psychotherapy might have some adverse effects in specific cases. It has been previously reported that excessive use of games led veterans to a lower perceived quality of life [95]. Excessive play can potentially lead to video game addiction, included as a behavioral addiction in the DSM-5 section recommending further research. According to several studies, gaming addiction has a higher association with other psychiatric disorders, such as affective disorders, depression, anxiety disorders, and ADHD, so these diagnoses should be taken into account as a potential risk factor. Furthermore, according to LC4MP and flow theory, cognitive overload may occur if a video game challenge is too high for the available skillset. In the long run, this may represent a dangerous condition for the patient, eliciting emotional and practical frustration. However, the advantages of video game play in psychotherapy generally outweigh the risks connected to excessive play [95], considering the protective role of the therapist in guiding the gameplay sessions. As an emotional coach, the therapist must guide the patient to assimilate video game play as an effective coping mechanism well integrated into everyday life. This paper aims to broaden VGT^®^ use and to make this approach more solid and standardized by defining guidelines established in collaboration with other professionals who use video games as a therapeutic tool in and outside Italy. Currently, experiences of VGT^®^ use are underway in three different associations. These include the Hub Costa Volpino (BG, Italy), where 20 young people are involved in a project on NEETs, and the Gaia Cooperative of Lumezzane (BS, Italy), which is a residential community for gambling addicts (DGA), where eight users use VGT^®^ for cognitive and emotional self-regulation. Both programs are weekly and are led by psychologists. Their focus is mainly on assessment (skill training) and insight, and, in the case of the Gaia Cooperative community, on storytelling as well. In addition, the programs are aimed at promoting Flow, starting from positive reinforcement, awareness of the characteristics of the Self that come alive in the video game, and observing the patients’ playful style (as considered by Briggs Type Indicator (MBTI) [29] and Bartle test [31]). A third center using VGT^®^ is the CAG Tok Tok Center in Travagliato (BS, Italy) of Fraternità Community. In this center, VGT^®^ is used for minor deviations and young people with court-ordered probation by the Juvenile Court of Brescia. The work carried out in this center involves five young people between the ages of 15 and 18 and focuses on cognitive and emotional self-regulation, insight, and creative function. Based on the growing body of literature and the encouraging results of these programs, introducing commercial video games in psychotherapy may be considered an effective integrated approach. However, according to a recent systematic review of the benefits of role-playing games in psychotherapy, only 16% of the studies were definable as experimental [138]. Along the same lines, another systematic review highlighted a lack of longitudinal studies assessing long-term effects [139]. Considering there is a lack of consensus on standards for proper evaluation of their effectiveness [140], there is a need for more research on therapy centered around video games. For future work, we expect to be able to expand the sample of people using this approach to carry out studies that validate its effectiveness from a statistical point of view.

## Figures and Tables

**Figure 1 healthcare-11-01767-f001:**
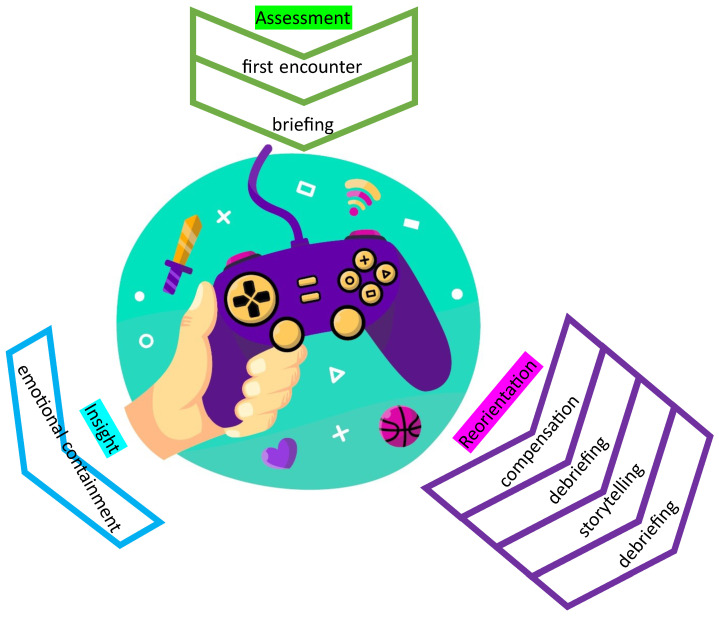
The different phases of VGT^®^.

**Table 1 healthcare-11-01767-t001:** Games and personality typologies.

Personality Typology	Game
**Analysts**	
Architect—INTJ	*Tetris;* *Monument Valley2; Thomas Was Alone; Inscryption*
Logician—INTP	*Dorfromantik; Don’t Starve Together; Security Booth: Director’s Cut; Green Heel*
Commander—ENTJ	*Valiant Hearths: The Great War; 35MM; Saturnalia; This War Of Mine*
Debater—ENTP	*The Last Campfire; Syberia: The World Before; How Fish Is Made; Mothered*
**Diplomats**	
Advocate—INFJ	*Tiny Lands; House Flipper; Townscraper; Death Stranding*
Mediator—INFP	*Night In The Woods; Coffee Talk; Life Is Strange; Detroit: Become Human*
Protagonist—ENFJ	*Overcooked: All You Can Eat; Among Us; Cult Of The Lamb; The Walking Dead: The Talltale Series*
Campaigner—ENFP	*Feather; Penguins Can Fly; Minecraft; My Summer Car*
**Sentinels**	
Logistician—ISTJ	*We Were Here Together; Pentiment; For Goodness Sake; She Sees Red: Interactive Thriller*
Defender—ISFJ	*ICO; Stray; It Takes Two; Little Misfortune*
Executive—ESTJ	*Oxygen Not Included; Yes, Your Grace; Papers, Please; Not For Broadcast*
Consul—ESFJ	*Animal Crossing: New Horizons; Hokko Life; The Last Guardian; A Normal Lost Phone*
**Explorers**	
Virtuoso—ISTP	*Freud’s Bones; Road96; Fear To Fathom: Home Alone; Scorn*
Adventurer—ISFP	*Tunic; Little Orpheus; Epystory—Typing Chronicles; The Pathless*
Entrepreneur—ESTP	*Horizon Chase 2; Getting Over It With Bennett Foddy; Cuphead; The Binding Of Isaac*
Entertainer—ESFP	*Untitled Goose Game; Crypt Of The Necromancer; Find The Murderer 2; Dude Simulator 3*

## Data Availability

Not applicable.

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
