# Peer review of "Putting the Gaming Experience at the Center of the Therapy—The Video Game Therapy® Approach"

_healthcare, 2023, doi:10.3390/healthcare11121767_

Round 1
Reviewer 1 Report
Manuscript Healthcare-2391291 on "Putting The Gaming Experience At The Center Of The Therapy - The Video Game Therapy Approach."
Video games are increasingly being used as therapy for various mental illnesses. Research has shown that video games can treat depression, anxiety, and addiction. One of the main benefits of video games in therapy is that they can provide a sense of engagement and immersion that traditional therapeutic methods may lack. The article proposes an approach, Video Game Therapy®, in which the game in which the gaming experience is placed at the center of therapy, as playing video games could facilitate patients to reach the conditions in which different therapeutic methodologies and approaches might work best. To improve their work, I suggest
1. Include in the summary section the most relevant results and a paragraph with the limitations of this research and future work.
2. Review the English grammar and spelling of the entire paper.
3. Apply the appropriate style of Table 1 for MDPI journals.
4. Improve the resolution and quality of Figure 1.
5. Summarize the method in an outline or concept map.
6. To make your research more impactful, place the study data and analysis in a data set; you can create one at https://data.mendeley.com/.
7. Strengthen the discussion section by comparing your contributions with those of other authors.
8. Strengthen the conclusion section.
9. Update the references of the 105 references; 88.5% are out of the five years of validity.
10. Apply the citation and referencing style of the journal; review https://www.mdpi.com/journal/healthcare/instructions.
Review the English grammar and spelling of the entire paper.
Reviewer 2 Report
This manuscript explores the therapeutic intervention theory and specific methods of video games for mental health. We know that the game science is an emerging and still partially undefined area of scientific research, characterized by a strong multidisciplinary focus spanning many different disciplines. The manuscript has sufficient evidence and detailed data, and has high practical application value.
This study mainly uses Adler psychology and MBTI personality analysis model as the theoretical framework of psychological treatment, and these contents should be reflected in the abstract.
There is still a lack of high-quality clinical research evidence for play therapy in psychotherapy. The text of this article should be more rational, and the use of affirmative (causal) language is not recommended.
In addition, play therapy may have some adverse effects, such as over-dependence, over-concentration, and other physiological health risks, which must be reflected in the conclusions.
Round 2
Reviewer 1 Report
The document has been significantly improved by applying the reviewers' comments.
I suggest considering it for publication.
The document has been significantly improved by applying the reviewers' comments.
I suggest considering it for publication.